# ELIMINATING THE FIRST MOMENT STATE IN ADAM OPTIMIZER

## ABSTRACT

The ADAM optimizer and its variants are widely used in large-scale machine learning, but their memory footprint is costly due to the storage of two state variables. In ADAM, the exponential moving average (EMA) of gradients ($m$) is used to estimate the first moment. We observe that although $m$ primarily serves as a first-moment estimator, it also carries variance information that can be exploited to estimate the second moment. Moreover, the gradient buffer can be repurposed to handle both gradient accumulation and a proxy for the first moment, effectively folding $m$ into the gradient buffer itself. These modifications reduce the number of optimizer state variables from two to one per parameter, yielding Half-Memory ADAM (HMADAM) and its decoupled-weight-decay variant (HMADAMW). This variant uses ADAM update rule and hyper-parameters. Experiments across discriminative and generative tasks including CNNs, transformers, and diffusion models show that HMADAMW matches the performance of standard ADAMW in convergence speed, final accuracy, and runtime, while substantially lowering memory usage. This makes it a practical choice for memory-constrained training scenarios such as large-scale language modeling.

## 1 INTRODUCTION

Most large-scale models are trained with the ADAM (Kingma & Ba, 2014) family of optimizers, such as ADAMW (Loshchilov & Hutter, 2019). While these optimizers are effective and widely adopted, their memory overhead poses a fundamental challenge for scaling. ADAM uses memory to store two auxiliary state variables per parameter (estimates of the first and second moments). For example, training DEEPSEEK-V3 with 671 billion parameters entails an additional 1.34 trillion optimizer state variables. Even in `bfloat16` precision, one replica corresponds to approximately 2.7 TB of optimizer state across devices. To accommodate this scale, DEEPSEEK-V3 offloads its optimizer states to CPU host memory, introducing communication overhead and slowing training. Thus, the cost of maintaining optimizer states has become a primary obstacle to scaling, motivating research into more memory-efficient optimization methods.

A broad literature explores memory-efficient optimization methods. Some techniques propose new optimizers that use fewer states (e.g., SWAN (Ma et al., 2025), MUON (Liu et al., 2025a), MNGD (Scetbon et al., 2025)), while others focus on making ADAM itself more memory-efficient. Approaches in the latter category generally compress the first and second moments, most often via quantization (Dettmers et al., 2022), low-rank approximation (Zhao et al., 2024a; Chen et al., 2024), or grouping parameters (Zhang et al., 2025b) (see Section 2 for more details). In contrast to compression techniques, we propose a method to remove one of the two state variables altogether.

Table 1: Per-parameter state variables when training with ADAM (Kingma & Ba, 2014).

| Symbol | Name | Update rule | Stored in |
|:---:|---|---|---|
| $w$ | Parameter value | Gradient descent | Model state |
| $g$ | Gradient accumulator | Accumulation and ZERO_GRAD | Model state |
| $m$ | Estimate of first moment | Exponential moving average of $g$ | Optimizer state |
| $v$ | Estimate of second moment | Exponential moving average of $g^2$ | Optimizer state |

Table 1 summarizes the four state variables maintained during training. Typically, $w$ and $g$ are stored in the model object, while $m$ and $v$ are stored in the optimizer object. Thus, $m$ and $v$ are considered

optimizer state variables. Importantly, gradients are produced during backpropagation regardless of the optimizer, and are therefore not classified as optimizer state.

Our technique has three elements:

**Estimating $v$ from $m$ rather than from $g$.** ADAM maintains two exponential moving averages (EMAs): one of the gradients $g$ to approximate the first moment $\mathbb{E}[g]$, and another of the squared gradients $g^2$ to approximate the second moment $\mathbb{E}[g^2]$. Because gradients are stochastic, $m$, the EMA of $g$, inevitably inherits variance. Prior work has often regarded this variance as undesirable, associating it with instability or reduced generalization performance (Zhou et al., 2021; Liu et al., 2020; Zhuang et al., 2020). Our key insight is that this same variance in $m$ contains statistical information about the second moment (see Section 4). We exploit this to construct an estimator of $\mathbb{E}[g^2]$ from $m$ rather than from $g$.

**Combining the roles of $m$ and $g$.** Once the second-moment estimate no longer requires raw gradients, the gradient accumulator can be extended to serve as both a gradient buffer and a proxy for the first-moment estimate. In most deep learning frameworks such as PYTORCH (Paszke et al., 2019), gradients are reset each step via the ZERO_GRAD operation, after which new gradients are accumulated into $g$. We replace this reset with a decay operation $g \leftarrow \beta_1 g$, so that $g$ evolves into an exponentially decayed accumulator across steps. We denote this decayed accumulator by $\tilde{g}$. By construction, $\tilde{g}$ serves two functions: (i) accumulating gradients within a training step, and (ii) maintaining an EMA across steps that can be used for the first-moment estimate. Moreover, $\tilde{g}$ can also be used in updating $v$. This dual use eliminates the need for a separate $m$ variable (see Section 3).

**Formal equivalence with ADAM through scaling adjustments.** The above two modifications provide a memory-efficient pathway to estimate the first and second moments of the gradient. Once these estimates are obtained, the parameter update rules remain the same as in ADAM, up to scaling adjustments that are derived and can be absorbed into the learning rate (see Section 4.1). We therefore present a variant of ADAM that preserves the update rules of ADAM while requiring one fewer auxiliary state variable per parameter. Extending this formulation with decoupled weight decay (Loshchilov & Hutter, 2019) yields Half-Memory ADAMW, which can serve as a drop-in replacement for ADAMW in existing training pipelines. Importantly, in our experiments, the learning rates tuned for ADAM and ADAMW transferred directly to HM-ADAM and HM-ADAMW. Our convergence analysis (Appendix D) and experiments (Section 5) show that convergence behavior and final accuracy are preserved across a range of architectures and tasks.

To experimentally evaluate our technique, we consider tasks ranging from small- to large-scale datasets and covering both discriminative and generative modeling (e.g., diffusion models and large language models), across vision (ImageNet) and NLP tasks (Transformers). Our experiments demonstrate that despite reducing one state variable, the convergence rate, final accuracy, and training speed remain statistically indistinguishable from the standard ADAM optimizer. Given the widespread use of ADAM, we believe this memory reduction can benefit a broad range of machine learning tasks, especially in training large language models where model and batch sizes continue to grow.

The rest of the paper is structured as follows. Section 2 reviews prior work and provides background for our method. Section 3 discusses how our approach applies to momentum SGD to save optimizer memory. Section 4 extends the method to the ADAM optimizer. Section 5 presents experimental results. Section 6 concludes with broader discussion and future directions. Additional derivations, pseudocode, and results are included in the appendix.

## 2 BACKGROUND

**Evolution of Optimizer State.** Early first-order optimizers such as vanilla stochastic gradient descent (SGD) did not maintain any optimizer state. Later, it was shown that tracking momentum with a single state variable per parameter (MOMENTUM SGD) significantly improved convergence (Polyak, 1964). Subsequently, the ADAM optimizer (Kingma & Ba, 2014) introduced a second state variable, yielding further improvements. Between 2015 and 2025, model sizes have grown by several orders of magnitude (Our World in Data, 2025), while memory capacity has increased far more modestly (McCallum, 2024). This widening disparity has made memory efficiency a central concern in large-scale training. Our technique builds on this trajectory by proposing a memory-efficient variant of ADAM.

## 2.1 MEMORY-EFFICIENT ADAM VARIANTS

Efforts to reduce the memory footprint of ADAM can be broadly grouped into two categories.

**State compression.** Quantization is the most widely studied approach: 8-bit ADAM retains 32-bit accuracy while significantly reducing memory across tasks such as language modeling and ImageNet classification (Dettmers et al., 2022; Wu et al., 2018; Liu et al., 2024). More aggressive 4-bit methods address outliers using finer block sizes and row/column normalization (Li et al., 2023; Wang et al., 2024b; Mahdavinia & Mahdavi, 2025), while 1-bit approaches such as SIGNSGD encode updates using only their sign (Bernstein et al., 2018). Beyond quantization, low-rank methods compress states with matrix approximations (Zhao et al., 2024b; Yen et al., 2023; Zhao et al., 2024a) or through low-rank parameter updates (Hu et al., 2022; Vogels et al., 2019; Zhang et al., 2024). GALORE reduces optimizer memory by projecting gradients into a low-rank subspace (Su et al., 2025; Glentis et al., 2025b; Zhang et al., 2025a), while FIRA augments GALORE with residual full-rank corrections for stronger performance (Zmushko et al., 2024). Further refinements include Q-GALORE, LDADAM, and BADAM (Su et al., 2025; Modoranu et al., 2025). Sparsity-based techniques enforce structure on gradients or states to reduce memory (Yang et al., 2024; Chen et al., 2020; Han et al., 2024), while structured adaptation methods such as APOLLO (Zhu et al., 2024) and SCALE (Glentis et al., 2025a) approximate per-parameter adaptivity with compressed statistics rather than full moments.

**State management.** Other techniques reduce memory pressure by reorganizing how states are stored or distributed. Partitioning and offloading systems such as ZERO (Rajbhandari et al., 2020), FSDP (Zhao et al., 2023), and DEEPOPT (Wang et al., 2024a) shard or move states to CPU or disk. Memory-efficient update rules such as SM3 (Anil et al., 2019) replace per-parameter statistics with row- or column-wise accumulators, reducing the dimensionality of stored state.

**Alternative optimizers.** A parallel line of work departs from ADAM's two-state design altogether. MUON emulates adaptive behavior by combining momentum with approximate orthogonalization, eliminating second-moment tracking (Jordan et al., 2024; Liu et al., 2025a). Hybrid approaches such as COSMOS (Liu et al., 2025b) split optimization across eigensubspaces, applying SOAP (Vyas et al., 2025) for leading modes and MUON for the remainder. Other approaches, such as ADAM-MINI (Zhang et al., 2025b; Kalra et al., 2025), restructure adaptivity by assigning block-partitioned learning rates based on spectral information. SWAN (Ma et al., 2025) eliminates optimizer states entirely by applying row-wise normalization and whitening, achieving the same memory footprint as plain SGD. Gradient Multi-Normalization (MNGD) generalizes this idea by combining multiple normalization schemes (Scetbon et al., 2025).

These methods demonstrate that large-scale training is possible without ADAM's full two-state design, but they introduce fundamentally new optimizers rather than modifications of ADAM. To our knowledge, no prior work has reduced ADAM's state count directly: existing methods either compress or redistribute its two auxiliary states. In contrast, our method halves ADAM's memory footprint by coupling the estimation of the first and second moments while preserving its update rule.

## 2.2 MEMORY SAVINGS BEYOND THE OPTIMIZER

Optimizer state is only one contributor to training memory; activations and attention key–value caches also consume substantial memory during forward and backward passes. These components are orthogonal to optimizer design and thus outside the direct scope of this work, but we briefly review them because their standard remedy, *gradient accumulation*, interacts with activation memory.

**Activation memory.** Backpropagation requires storing activations from the forward pass. Gradient checkpointing (Chen et al., 2016) reduces this cost by selectively recomputing intermediates during the backward pass. Transformer-specific techniques such as FLASHATTENTION (Dao et al., 2022), XFORMERS (Research, 2022), MULTI-QUERY ATTENTION (Shazeer, 2019), and PAGEDATTENTION (Chen et al., 2023b) further reduce memory by optimizing attention activations and key–value caches. A complementary strategy is *micro-batching*, which splits a mini-batch into sequential micro-batches, lowering peak activation memory while preserving the effective batch size.

| **Algorithm 1** Standard MOMENTUM SGD | **Algorithm 2** Stateless MOMENTUM SGD |
|---|---|
| **Require:** Learning rate $\eta$ | **Require:** Learning rate $\eta$ |
| **Require:** Decay rate $\beta_1$ | **Require:** Decay rate $\beta_1$ |
| **Require:** Number of steps $T$ | **Require:** Number of steps $T$ |
| 1: $m_0 \leftarrow 0$ | 1: $\tilde{g}_0 \leftarrow 0$ |
| 2: **for** $t = 1$ to $T$ **do** | 2: **for** $t = 1$ to $T$ **do** |
| 3: $\quad g_t \leftarrow 0$ ▷ ZERO_GRAD | 3: $\quad \tilde{g}_t \leftarrow \beta_1 \tilde{g}_{t-1}$ ▷ DECAY_GRAD |
| 4: $\quad g_t \leftarrow g_t + \nabla w_t$ | 4: $\quad \tilde{g}_t \leftarrow \tilde{g}_t + \nabla w_t$ |
| 5: $\quad m_t \leftarrow \beta_1 m_{t-1} + (1-\beta_1)g_t$ | 5: |
| 6: $\quad w_t \leftarrow w_{t-1} - \eta m_t$ | 6: $\quad w_t \leftarrow w_{t-1} - \eta(1-\beta_1)\tilde{g}_t$ |
| 7: **end for** | 7: **end for** |

Figure 1: Comparison between MOMENTUM SGD and STATELESS MOMENTUM SGD. In the stateless version, line 1 initializes $\tilde{g}_0$ instead of $m_0$, line 2 replaces ZERO_GRAD with DECAY_GRAD, and line 4 removes the need to store $m_t$. Line 5 applies the correction factor $(1-\beta_1)$ to $\tilde{g}_t$ before the update. Importantly, $\tilde{g}_t$ resides in the model's gradient buffer, which is required by backpropagation regardless of optimizer. Thus, from the optimizer's perspective, this memory comes at no extra cost.

## 3 REUSING GRADIENT ACCUMULATOR TO STORE THE FIRST MOMENT

In micro-batching, each micro-batch contributes gradients that are added into the gradient accumulator $g$ for every parameter $w$. Conventionally, this buffer is cleared at the start of each training step using ZERO_GRAD; after all micro-batches have been processed, the accumulated $g$ is used to update the parameters. This practice discards potentially useful information at every step. We propose replacing the reset with a decay update ($g \leftarrow \beta_1 g$), so that the gradient buffer itself evolves into an exponential moving average (EMA) across steps.

**Derivation.** Let $m_t$ denote the EMA of gradients at step $t$ with decay $\beta_1$:

$$m_t = \beta_1 m_{t-1} + (1-\beta_1)g_t, \quad m_{-1} = 0. \tag{1}$$

Define the rescaled variable $\tilde{g}_t = \frac{1}{1-\beta_1}m_t$. Substituting into Eq. 1 gives

$$(1-\beta_1)\tilde{g}_t = \beta_1(1-\beta_1)\tilde{g}_{t-1} + (1-\beta_1)g_t,$$

which simplifies to

$$\tilde{g}_t = \beta_1 \tilde{g}_{t-1} + g_t, \tag{2}$$

with $\tilde{g}_{-1} = 0$.

Equation 2 shows that $\tilde{g}_t$ can be updated with only two in-place operations: (i) decay the buffer by $\beta_1$, and (ii) add the current gradient $g_t$. If we store $\tilde{g}_t$ directly in the gradient buffer already allocated by the framework, then $m_t = (1-\beta_1)\tilde{g}_t$ is *numerically identical* to the conventional EMA in Eq. 1. In other words, the first-moment estimate can be recovered without explicitly storing $m_t$ as an additional optimizer state variable. The only modification is to replace ZERO_GRAD with a decay operation.

This reuse is both memory-neutral (no new buffers are created) and exact (the recovered $m_t$ matches standard momentum). Conceptually, this reframes the gradient buffer from a short-lived scratch space into a persistent accumulator of information across steps. A similar observation was briefly noted in a footnote by Sohoni et al. (2019) for MOMENTUM SGD, and gradient decay also appears in the design of MNGD (Scetbon et al., 2025). However, these works neither formalized the equivalence nor applied it in the context of ADAM. Our contribution is to generalize this reinterpretation, showing that buffer reuse with decay allows the first-moment state in ADAM to be eliminated entirely, while preserving ADAM's update rule.

## 4 HALF-MEMORY ADAM

In standard ADAM, the second-moment accumulator $v_t$ is updated directly from the raw gradients $g_t$, which presumes storing $g_t$ at each iteration. In our variant, only the exponentially weighted average $\tilde{g}_t$ is maintained. The central question is whether the information required for updating $v_t$ can be recovered from $\tilde{g}_t$ alone.

**Algorithm 3** Original ADAM optimizer

**Require:** Learning rate $\eta$
**Require:** Decay rates $\beta_1, \beta_2$
**Require:** Numerical constant $\epsilon$
**Require:** Number of steps $T$
1:   $m_0 \leftarrow 0, v_0 \leftarrow 0$        ▷ Initialize moments
2: **for** $t = 1$ to $T$ **do**
3:     $g_t \leftarrow 0$                 ▷ ZERO_GRAD
4:     $g_t \leftarrow g_t + \nabla w_t$     ▷ Accumulate gradient
5:     $m_t \leftarrow \beta_1 m_{t-1} + (1 - \beta_1) g_t$ ▷ First moment
6:     $v_t \leftarrow \beta_2 v_{t-1} + (1 - \beta_2) g_t^2$ ▷ Second moment
7:     $\hat{m}_t \leftarrow m_t / (1 - \beta_1^t)$       ▷ Bias correction
8:     $\hat{v}_t \leftarrow v_t / (1 - \beta_2^t)$
9:     $w_t \leftarrow w_{t-1} - \eta \cdot \hat{m}_t / (\sqrt{\hat{v}_t} + \epsilon)$    ▷ Update
10: **end for**

**Algorithm 4** Half-Memory ADAM optimizer

**Require:** Learning rate $\eta$
**Require:** Decay rates $\beta_1, \beta_2$
**Require:** Numerical constant $\epsilon$
**Require:** Number of steps $T$
1:   $\tilde{g}_0 \leftarrow 0, v_0 \leftarrow 0$   ▷ $m$ is no longer a state variable
2: **for** $t = 1$ to $T$ **do**
3:     $\tilde{g}_t \leftarrow \beta_1 \tilde{g}_{t-1}$             ▷ DECAY_GRAD
4:     $\tilde{g}_t \leftarrow \tilde{g}_t + \nabla w_t$
5:     $m_t \leftarrow (1 - \beta_1) \tilde{g}_t$
6:     $v_t \leftarrow \beta_2 v_{t-1} + (1 - \beta_2) \tilde{g}_t^2 \times (1 - \beta_1^2)$
7:     $\hat{m}_t \leftarrow m_t / (1 - \beta_1^t)$
8:     $\hat{v}_t \leftarrow v_t / (1 - \beta_2^t)$
9:     $w_t \leftarrow w_{t-1} - \eta \cdot \hat{m}_t / (\sqrt{\hat{v}_t} + \epsilon)$
10: **end for**

Figure 2: Comparison between the original ADAM optimizer and HALF-MEMORY ADAM. Colored terms indicate differences: red marks the original, green marks the half-memory variant. For clarity, changes between $g_t$ and $\tilde{g}_t$ are not highlighted. Key differences are: (1) $m_0$ is no longer initialized or stored (line 1); (2) the gradient reset (ZERO_GRAD) is replaced by DECAY_GRAD (line 3); (3) $m_t$ is not updated recursively but recovered from $\tilde{g}_t$ (line 5); and (4) the correction factor $(1 - \beta_1^2)$ is applied to $v_t$ to adjust for the new estimation (line 6).

**Moments as statistical summaries.** First- and second-moment accumulators are statistical summaries, not uniquely defined estimators. Within the ADAM family, different decay parameters $(\beta_1, \beta_2)$ correspond to different effective averaging windows, yet all are valid estimators of the same underlying quantities. What defines the family is the update rule rather than the precise estimator. Consequently, constant factors arising from alternative estimators can be absorbed into the stepsize schedule without altering the algorithm's essential behavior.

**Intuition.** The stochastic gradient $g_t$ can be modeled as a random variable with mean $\mu_t$ and variance $\sigma_t^2$. A substantial body of prior work has shown that in practical training regimes the mean is typically much smaller than the standard deviation ($\mu_t \ll \sigma_t$), a property commonly described as the low signal-to-noise ratio (SNR) of gradients. Although often regarded as a challenge for optimization, this property can be leveraged to our advantage. In particular, when $\mu_t \ll \sigma_t$, the second moment of the raw gradient and that of its exponential moving average $\tilde{g}_t$ become approximately proportional:

$$\mathbb{E}[g_t^2] \approx (1 - \beta_1^2) \mathbb{E}[\tilde{g}_t^2]. \tag{3}$$

Intuitively, $\mathbb{E}[g_t^2]$ measures the overall scale of gradient fluctuations. If this quantity is large, then $\mathbb{E}[\tilde{g}_t^2]$ will also be large, since $\tilde{g}_t$ is a weighted average of past gradients and thus preserves the same variance structure.

The formal derivation of Equation 3 is given in Appendix A. This derivation relies on the fact that in practice $\text{SNR}_t^2 = \mu_t^2 / \sigma_t^2 \to 0$. Both theoretical analyses (Mandt et al., 2017; Balles & Hennig, 2018; McCandlish et al., 2018) and our empirical measurements (Appendix B) confirm this behavior, with typical SNR values around $10^{-2}$–$10^{-3}$. Empirically, Figure 8 (left) shows that $\mu^2 / \sigma^2$ decreases during training. Figure 8 (right) compares the left- and right-hand sides of Equation 3; the points lie tightly along the diagonal (Pearson correlation $= 0.99$), demonstrating that $\tilde{g}_t$ reliably preserves the variance structure of $g_t$. Together, these results justify the use of equation 3 in our optimizer.

### 4.1 BACKWARD COMPATIBILITY WITH ADAM

The pseudocode of the original ADAM and the HALF-MEMORY ADAM are compared in Figure 2. After we estimate the first and second moments using $\tilde{g}_t$, we apply the same bias-correction procedure as in the original ADAM, and perform the parameter update identically (lines 7–9). ADAMW can be obtained by adding a single weight decay line at the start of the pseudocode; we omit a separate listing for brevity, as the modification is minimal.

While the pseudocode illustrates the core algorithm, we introduce two provisions to improve runtime efficiency and ensure backward compatibility with existing workflows.

**Absorbing scaling factors into the learning rate.** The scaling factors used in our moment computations remain constant during training and can be absorbed into the learning rate to avoid redundant operations. Specifically, the factors $(1 - \beta_1)$ and $(1 - \beta_1^2)$ can be combined into a single rescaling. Applying the following transformation:

$$\eta' = \eta \cdot \sqrt{\tfrac{1-\beta_1}{1+\beta_1}}, \tag{4}$$

preserves the numerical behavior of the original ADAM update while avoiding extra multiplications. This adjustment keeps the implementation both efficient and equivalent. The derivation and a modified pseudocode are provided in Appendix C.

**Backward-compatible integration via ZERO_GRAD override.** To maintain compatibility with standard PYTORCH workflows, we override the `zero_grad` method in our optimizer classes. This redirects gradient-reset behavior to our `decay_grad` operation, allowing the gradient buffer to maintain the EMA form $\tilde{g}$ without requiring user-side changes. As a result, switching to our memory-efficient optimizer requires only replacing `Adam` or `AdamW` with `HMAdam` or `HMAdamW`, with no modifications to training loops or gradient handling logic. The implementation remains fully compatible with autograd, gradient accumulation, and distributed training.

## 5 EXPERIMENTS

**Objective.** The primary objective of our work is to reduce the optimizer state memory required by ADAM. The central question is whether this memory reduction can be achieved without altering training dynamics. Accordingly, our experiments are designed to test whether HMADAMW exhibits the same convergence behavior, and runtime efficiency as the standard ADAMW. In addition to optimizer memory usage, we compare training loss and total training time.

**Summary.** Across all benchmarks, the optimizer state memory is reduced by exactly half, as predicted by our design. Moreover, given identical hyperparameters, the convergence curves of HMADAMW closely match those of ADAMW. When similar hyperparameters (learning rate, $\beta_1$, $\beta_2$, $epsilon$) are given to HMADAMW and ADAMW, their training loss and validation accuracy are statistically indistinguishable, and wall-clock training time is unaffected. These results confirm that the proposed optimizer is a memory-efficient drop-in replacement rather than a new optimization method.

**Benchmarks.** We evaluate four representative tasks:

1. **MNIST** (LeCun et al., 1998) classification with a ConvNet, to systematically analyze sensitivity to optimizer hyperparameters.

2. **Autoregressive language modeling** on the C4 dataset (Raffel et al., 2020) using LLAMA models ranging from 150M to 3B parameters, to test scalability and stability in large-scale pretraining.

3. **Image generation** on the CelebA dataset (Liu et al., 2015) with a UNET-based diffusion model (Ho et al., 2020), to evaluate generative modeling performance.

4. **ImageNet** (Deng et al., 2009) classification with EFFICIENTNET-B7 (Tan & Le, 2019), to assess behavior in a high-capacity convolutional network. Results for image generation and ImageNet are reported in Appendix E.

**Hardware.** LLM experiments were conducted on NVIDIA H100 GPUs with 80 GB of memory; all other experiments ran on an NVIDIA Quadro RTX 6000 (24 GB). In LLAMA experiments, both parameters and optimizer states were stored in `bfloat16`; for all other tasks, we used `fp32`. Batch sizes and other experimental details are specified in the respective subsections below.

### 5.1 MNIST: SIMILAR BEHAVIOR GIVEN SIMILAR HYPERPARAMETERS

We use the MNIST dataset with a simple ConvNet architecture to study the effect of hyperparameter choices in a fast, controlled setting. Specifically, we adopt the PYTORCH reference ConvNet

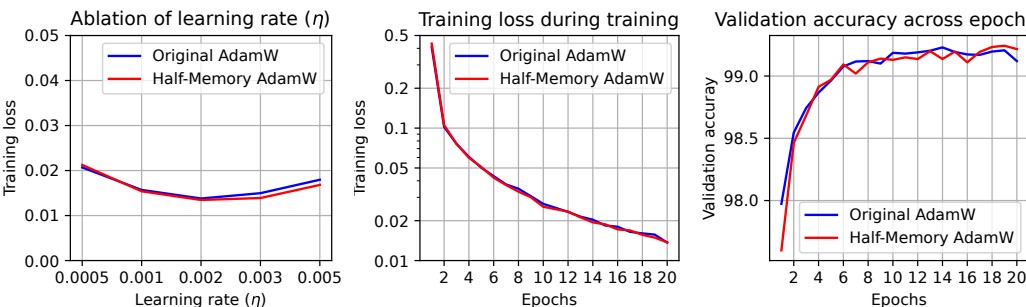

Figure 3: Comparison between the original ADAM optimizer and the proposed HALF-MEMORY ADAM on MNIST classification. **Left:** Final training loss after 20 epochs across a range of learning rates. For each optimizer, the best-performing $(\beta_1, \beta_2)$ combination was selected from the grid in Figure 4. The curves show that the choice of learning rate $\eta$ has a much larger effect on final loss than the choice of optimizer. **Middle:** Training loss over 20 epochs using the best $(\eta, \beta_1, \beta_2)$ configuration for each optimizer. The curves largely overlap, indicating that both optimizers follow similar optimization trajectories. **Right:** Validation accuracy throughout training, showing statistically indistinguishable generalization behavior.

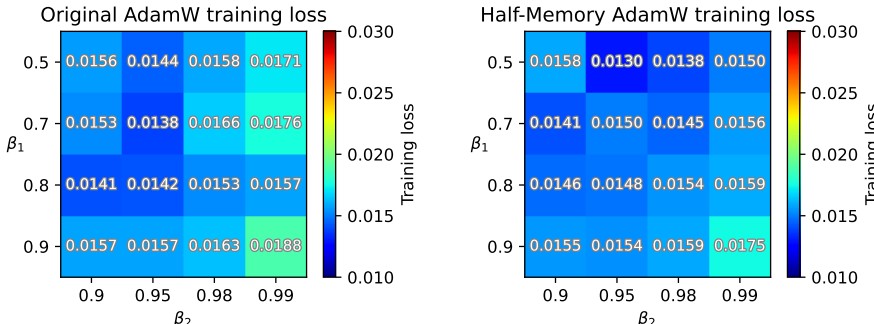

Figure 4: Final training loss after 20 epochs for different $(\beta_1, \beta_2)$ settings on MNIST classification, with a fixed learning rate of 0.002. **Left:** ADAMW. **Right:** HALF-MEMORY ADAMW. The grid of $(\beta_1, \beta_2)$ values shown here was used in Figure 3 to select the best hyperparameters for each learning rate. As shown, the choice of $(\beta_1, \beta_2)$ has a larger impact on final loss than the choice of optimizer.

implementation (PyTorch Team, 2023), which consists of two convolutional layers followed by two fully connected layers with ReLU activations and dropout, totaling approximately 1.2M parameters.

We compare ADAMW and HMADAMW across 80 hyperparameter configurations, defined as the Cartesian product of 5 learning rates, 4 values of $\beta_1$, and 4 values of $\beta_2$ (Figures 3 and 4). Each configuration is trained for 20 epochs with a batch size of 1000 and repeated three times for averaging. Figure 3 shows that both optimizers respond similarly to learning rate variations, exhibit overlapping training loss trajectories, and achieve comparable validation accuracy. Figure 4 highlights the effect of $(\beta_1, \beta_2)$ combinations, demonstrating that the two optimizers behave statistically indistinguishably under matched hyperparameters.

We also measured memory usage and runtime. In our experiments, model parameters and gradients each used 4.8 MB of memory. ADAMW required an additional 9.6 MB for optimizer state, while HMADAMW required only 4.8 MB, as predicted. Wall-clock measurements across 480 training runs showed that ADAMW averaged 0.36 ms per training step, while HMADAMW averaged 0.34 ms. The remainder of the training iteration, including data loading and forward/backward passes, took 2.21 ms in both cases, indicating comparable runtime and a slight improvement for HMADAMW due to reduced memory overhead.

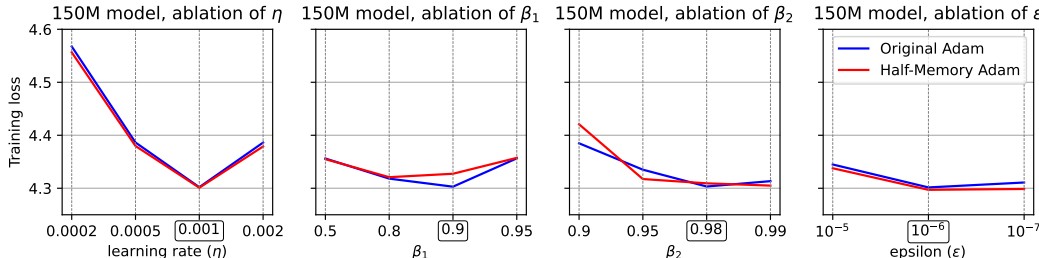

Figure 5: Ablation of optimizer hyperparameters using a 150M-parameter LLaMA3-style language model trained on 256 million tokens. Each subplot varies a single hyperparameter—$\eta$, $\beta_1$, $\beta_2$, or $\epsilon$—while keeping the others fixed at their default values: $\eta = 0.001$, $\beta_1 = 0.9$, $\beta_2 = 0.98$, and $\epsilon = 10^{-6}$. These default settings are highlighted using boxed tick labels in each subplot. The y-axis (training loss) is shown with the same range across all four plots to allow direct visual comparison. Although each configuration was evaluated with a single training run, the results show consistent trends: the default setting yields the lowest training loss, and both Original Adam and Half-Memory Adam respond similarly to changes in each hyperparameter.

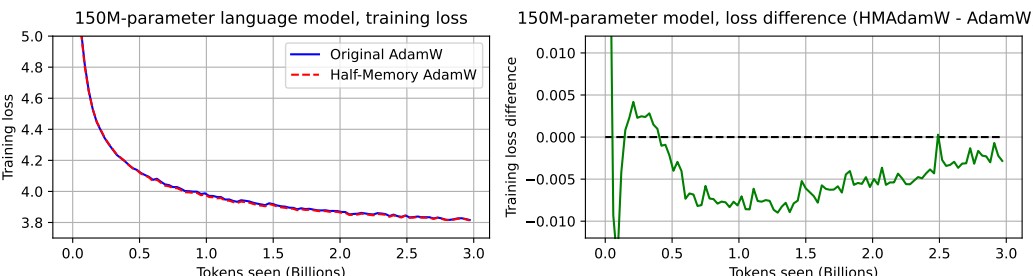

Figure 6: Comparison of training dynamics between the Original Adam and Half-Memory Adam optimizers using a 150M-parameter LLaMA-style language model trained on 3 billion tokens (20× the number of model parameters). **Left:** Training loss as a function of tokens seen. The two curves closely overlap, indicating that both optimizers follow nearly identical learning trajectories when trained under the same configuration. **Right:** Pointwise difference between the training losses of the two optimizers. The difference remains centered around zero and consistently within a margin of 0.01 throughout training.

## 5.2 LARGE LANGUAGE MODELS

Optimizer state memory is a major bottleneck in training large transformer-based language models, particularly at the billion-parameter scale. To evaluate the effectiveness of HMADAMW in this setting, we conduct experiments on LLaMA3-style models (Touvron et al., 2024) trained on the C4 dataset (Raffel et al., 2020) with a sequence length of 4096. All models are trained with next-token prediction and cross-entropy loss, using the standard LLaMA3 tokenizer and vocabulary. We use fixed learning rates without warmup or decay, store all state variables in `bf16`, and set the random seed to 0 for reproducibility.

We evaluate three model sizes—150M, 1B, and 3B parameters. The 150M model is used primarily to study sensitivity to hyperparameters, while the larger models test behavior at scale. For the 150M and 1B models, we use a batch size of 64K tokens, and for the 3B model we use 128K tokens. Figure 5 shows ablations over the four core ADAMW hyperparameters ($\eta$, $\beta_1$, $\beta_2$, and $\epsilon$) on the 150M model trained on 256M tokens. Both optimizers respond similarly across variations, with the default setting yielding the lowest training loss. Importantly, the training loss is statistically indistinguishable given identical hyperparameters. Figure 6 compares training trajectories for the same model over 3B tokens. The two optimizers closely align, with the difference in loss consistently below 0.01. Figure 5.1 extends the evaluation to larger models: a 1B model trained on 3B tokens, and a 3B model trained on 20B tokens. In both cases, HMADAMW mirrors the behavior of ADAMW, showing minimal divergence across learning rates and training curves. Together, these results demonstrate that HMADAMW preserves the convergence behavior of the original ADAMW across a wide range of hyperparameters and model sizes, while reducing memory usage.

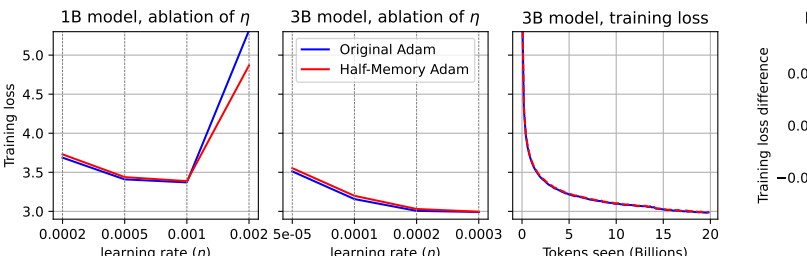

Figure 7: Comparison of ADAM and HALF-MEMORY ADAM on large-scale language modeling. **Left:** Learning-rate ablation on a 1B-parameter model trained on 3B tokens. **Center Left:** Learning-rate ablation on a 3B-parameter LLaMA3-style model trained on 20B tokens. **Center Right:** Training loss progression for the 3B model, showing nearly identical trajectories. **Right:** Pointwise loss difference from the previous plot, confirming the gap remains consistently small.

We also measured memory consumption. For the 3B model—the largest model we tested—ADAMW required 12 GB for its state variables, whereas HMADAMW required only 6 GB, cutting optimizer state memory in half and saving 6 GB overall. With a batch size of 128K tokens, both optimizers required 1.16 s per training step. The optimizer itself took on average 19 ms for ADAMW and 18 ms for HMADAMW. Each of the eight training runs for the 3B model took nearly 51 hours on a single H100 GPU, totaling about 400 GPU hours.

**Limitations.** While it would be desirable to evaluate even larger models (e.g., 8B–14B parameters), this was not feasible under our compute budget. In practice, training with standard ADAMW on a single 80 GB H100 GPU crashes due to memory exhaustion beyond 3B parameters, and running the 3B model already required nearly 400 GPU hours. Nevertheless, our results span two orders of magnitude in model size (150M to 3B parameters) and consistently show that HMADAMW matches the behavior of ADAMW. Since the memory savings arise from a structural reduction in optimizer state rather than scale-specific heuristics, we expect these findings to generalize to even larger models.

## 6 DISCUSSION

Our proposed HALF-MEMORY ADAM has three main strengths: (1) via an analytically derived scaling, the same hyperparameters that work for ADAM transfer directly to our optimizer (confirmed by our experiments); (2) our analysis indicates it retains ADAM's convergence characteristics under standard assumptions; and (3) it halves optimizer-state memory. In practice, swapping ADAM(W) for HMADAM(W) is a drop-in change: we override `zero_grad` with a decay step while all other hyperparameters and training code remain unchanged.

**Gradient clipping.** Not storing raw gradients does not preclude clipping or conditional skipping. With gradient accumulation, micro-batch gradients are transient in any case; frameworks expose them via backward hooks, enabling micro-batch–level clipping, which is often preferable to mini-batch–level clipping and remains fully compatible with our optimizer.

**Tuning of baseline.** A common concern in optimizer research is fair comparison, especially whether baselines are sufficiently tuned. To address this, we apply the same ablation strategy to both optimizers and use well-established hyperparameters known to perform well for ADAMW (e.g., $\beta_1 = 0.9$, $\beta_2 = 0.98$, $\epsilon = 10^{-6}$ in LLM training).

**Outlook and future work.** It would be valuable to explore how Half-Memory interacts with other memory-saving techniques, such as optimizer state quantization (Dettmers et al., 2022) and low-rank factorization (Dettmers et al., 2023), which may yield compounded efficiency gains. Second, our approach can be extended to other optimizers that rely on first-moment estimates including SIGNSGD (Bernstein et al., 2018), LION (Chen et al., 2023a), and LAMB (You et al., 2019).

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

## A  DERIVING THE SECOND MOMENT OF $g$ FROM THE SECOND MOMENT OF $\tilde{g}$

Consider the stochastic process of updating our new gradient accumulator $\tilde{g}$:

$$\tilde{g}_t = \beta_1 \tilde{g}_{t-1} + g_t, \quad g_t \sim \mathcal{N}(\mu, \sigma^2).$$

At steady-state equilibrium, expectations stabilize. First, we find the expectation explicitly:

$$E[\tilde{g}_t] = \beta_1 E[\tilde{g}_{t-1}] + \mu, \quad \Rightarrow \quad E[\tilde{g}] = \frac{\mu}{1 - \beta_1}.$$

Next, we compute the second moment at steady state:

$$E[\tilde{g}_t^2] = E[(\beta_1 \tilde{g}_{t-1} + g_t)^2] = \beta_1^2 E[\tilde{g}_{t-1}^2] + E[g_t^2] + 2\beta_1 E[\tilde{g}_{t-1} g_t].$$

Expanding the cross-term explicitly, we get:

$$E[\tilde{g}_{t-1} g_t] = E[\tilde{g}_{t-1}(\mu + (g_t - \mu))] = \mu E[\tilde{g}_{t-1}] + E[\tilde{g}_{t-1}(g_t - \mu)].$$

Since $\tilde{g}_{t-1}$ is determined only by gradients at previous steps (independent of the current centered noise $g_t - \mu$), we explicitly have:

$$E[\tilde{g}_{t-1}(g_t - \mu)] = 0.$$

Thus, the cross-term simplifies to:

$$E[\tilde{g}_{t-1} g_t] = \mu E[\tilde{g}_{t-1}] = \frac{\mu^2}{1 - \beta_1}.$$

Hence, at equilibrium:

$$E[\tilde{g}^2] = \beta_1^2 E[\tilde{g}^2] + E[g_t^2] + 2\beta_1 \frac{\mu^2}{1 - \beta_1}.$$

Solving explicitly for $E[g_t^2]$, we obtain the general result:

$$E[g_t^2] = (1 - \beta_1^2) E[\tilde{g}_t^2] - \frac{2\beta_1 \mu^2}{1 - \beta_1}. \tag{5}$$

In Appendix B we discuss the low SNR characteristics of gradients which implies $\mu \ll \sigma$. This implies:

$$\frac{\mu^2}{\sigma^2} \to 0.$$

Under this condition, the cross-term involving $\mu^2$ becomes negligible compared to the variance-dominated terms $E[g_t^2]$ and $\beta_1^2 E[\tilde{g}_{t-1}^2]$. Thus, Equation 5 simplifies to:

$$E[g_t^2] = (1 - \beta_1^2) E[\tilde{g}_t^2]. \tag{6}$$

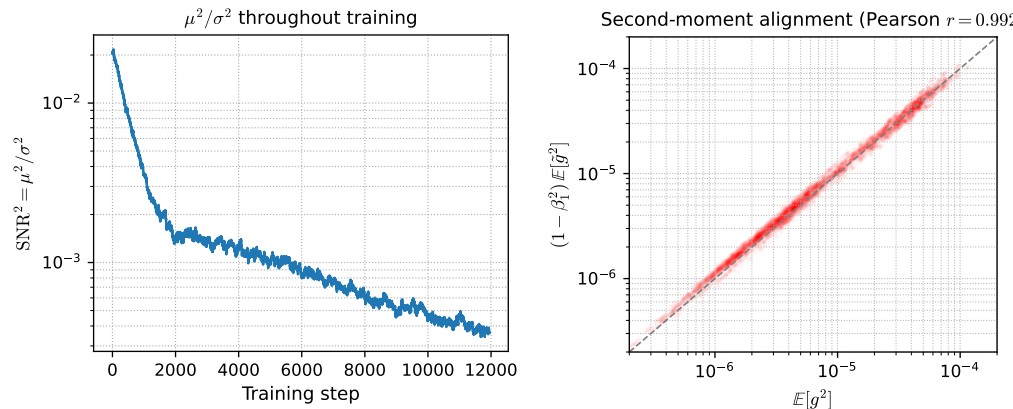

Figure 8: Empirical evidence supporting low signal-to-noise ratio. Left: Gradient $\mu^2/\sigma^2$ ratio decays as training progresses. **Right:** The second moment of $\tilde{g}$ strongly correlates with that of $g$, confirming that variance information is preserved.

## B   EMPIRICAL JUSTIFICATION FOR LOW SIGNAL-TO-NOISE RATIO

**Mathematical intuition.**   Let $g_t \in \mathbb{R}^d$ denote the stochastic gradient at step $t$, and let $\mu_t = \mathbb{E}[g_t]$ and $\sigma_t^2 = \mathrm{Var}(g_t)$ denote its mean and variance. At a local minimum $w^\star$, we have $\nabla f(w^\star) = 0$, so that $\mu_t \to 0$ as $w_t \to w^\star$. However, even in this regime, $\sigma_t^2$ remains nonzero due to sampling noise from mini-batches. Hence, the signal-to-noise ratio,

$$\mathrm{SNR}_t \;=\; \frac{\|\mu_t\|}{\sqrt{\mathbb{E}\|g_t - \mu_t\|^2}},$$

necessarily tends to zero as training progresses. This suggests that $\mu_t$ is asymptotically dominated by $\sigma_t$.

**Evidence from prior literature.**   The above intuition has been formalized and empirically validated in several works. Mandt et al. (2017) modeled SGD as approximate Bayesian inference, showing that in the stationary regime the mean vanishes relative to the variance, i.e. $\mathrm{SNR}_t \to 0$. Balles & Hennig (2018) reported that across CNN and LSTM layers, the variance exceeds the squared mean by orders of magnitude, yielding $\mathrm{SNR}_t < 10^{-2}$. McCandlish et al. (2018) introduced the concept of the *noise scale*, demonstrating that in large-batch language model training, gradient noise is typically $10^2$–$10^3$ times larger than the signal, i.e. $\mathrm{SNR} \approx 10^{-2}$–$10^{-3}$.

**Our empirical analysis.**   We validated these observations through experiments on the first 12,000 training steps of a 1 Billion parameter LLaMA model. Figure 8 reports two complementary findings. First, the ratio $\mu^2/\sigma^2$ decreases steadily as optimization progresses, confirming the trend documented in prior work and indicating that the signal-to-noise ratio diminishes throughout training. Second, the second moment of $\tilde{g}_t$ exhibits extremely high correlation with the raw second moment of $g_t$, demonstrating that $\tilde{g}_t$ faithfully preserves the variance structure of the underlying gradients. Together, these results provide strong empirical support for the validity of estimating second-moment statistics from $\tilde{g}_t$ alone.

Together, these results establish both theoretically and empirically that $\mu \ll \sigma$ in practical training regimes, and that $\tilde{g}_t$ is a faithful proxy for $g_t$ in terms of second-moment estimation. This justifies the design choice of estimating the second moment from $\tilde{g}_t$ in our optimizer.

---

**Algorithm 5** Half-Memory Adam Optimizer with Scaled Learning Rate

---

**Require:** Learning rate $\eta$
**Require:** Decay rates $\beta_1, \beta_2$
**Require:** Numerical constant $\epsilon$
**Require:** Number of steps $T$

1: $\eta' \leftarrow \eta \cdot \sqrt{\frac{1-\beta_1}{1+\beta_1}}$      ▷ Absorbed learning rate
2: $\tilde{g}_0 \leftarrow 0, v_0 \leftarrow 0$      ▷ $m$ is not stored
3: **for** $t = 1$ to $T$ **do**
4:     $\tilde{g}_t \leftarrow \beta_1 \tilde{g}_{t-1}$      ▷ DECAY_GRAD
5:     $\tilde{g}_t \leftarrow \tilde{g}_t + \nabla w_t$      ▷ gradient accumulation
6:          ▷ Empty because $\tilde{g}_t$ replaces $m_t$
7:     $v_t \leftarrow \beta_2 v_{t-1} + (1-\beta_2)\tilde{g}_t^2$      ▷ unchanged
8:     $\hat{m}_t \leftarrow \tilde{g}_t/(1-\beta_1^t)$      ▷ $\tilde{g}_t$ replaces $m_t$
9:     $\hat{v}_t \leftarrow v_t/(1-\beta_2^t)$      ▷ unchanged
10:    $w_t \leftarrow w_{t-1} - \eta' \cdot \hat{m}_t/(\sqrt{\hat{v}_t} + \epsilon)$      ▷ New learning rate is used.
11: **end for**

---

Figure 9: More efficient implementation of Half-Memory Adam Optimizer. For efficiency, the factors of $m$ and $v$ are absorbed to a new learning rate. The differences with the original For clarity, differences with Adam optimizer algorithm 3 are highlighted in green. ADAMW pseudo-code needs one extra line to apply weight decay in the beginning of the loop. We omitted it to avoid repetition.

## C    ABSORBING SCALING FACTORS INTO THE LEARNING RATE

To avoid redundant runtime multiplications and make our optimizer update rule algebraically identical to the original Adam formulation, we can absorb constant scaling factors into the learning rate.

Our method uses $\tilde{g}_t$, the exponential moving average (EMA) of the gradient $g_t$, in place of $g_t$. The first and second moments are then computed as:

$$m_t = (1-\beta_1)\tilde{g}_t, \tag{7}$$

$$v_t = \beta_2 v_{t-1} + (1-\beta_2)(1-\beta_1^2)\tilde{g}_t^2. \tag{8}$$

The bias-corrected forms are:

$$\hat{m}_t = \frac{m_t}{1-\beta_1^t}, \tag{9}$$

$$\hat{v}_t = \frac{v_t}{1-\beta_2^t}. \tag{10}$$

Therefore, the update becomes:

$$w_t \leftarrow w_{t-1} - \eta \cdot \frac{\hat{m}_t}{\sqrt{\hat{v}_t} + \epsilon}.$$

Ignoring $\epsilon$ for simplicity, the expression becomes:

$$w_t \leftarrow w_{t-1} - \eta \cdot \frac{(1-\beta_1)\tilde{g}_t}{\sqrt{(1-\beta_1^2)\tilde{g}_t^2}} = w_{t-1} - \eta \cdot \frac{1-\beta_1}{\sqrt{1-\beta_1^2}} \cdot \frac{\tilde{g}_t}{|\tilde{g}_t|}.$$

This shows that our update differs from the original Adam update only by a constant multiplicative factor:

$$\eta' = \eta \cdot \frac{1-\beta_1}{\sqrt{1-\beta_1^2}} = \eta \cdot \sqrt{\frac{1-\beta_1}{1+\beta_1}}.$$

Thus, to exactly recover the behavior of the original Adam optimizer, we can replace the learning rate $\eta$ with

$$\eta' = \eta \cdot \sqrt{\frac{1-\beta_1}{1+\beta_1}}.$$

We can apply the same factor to epsilon ($\epsilon$) as well. This substitution ensures that no runtime changes in numerical behavior occur and that the update rule is algebraically identical to Adam's. In practice, we pre-scale the learning rate accordingly to avoid introducing additional multiplications into the training loop. The pseudo-code of the absorbed learning-rate implementation is illustrated in Figure 9.

## D  CONVERGENCE CHARACTERISTICS OF OUR ADAM VARIANT

It is well known that standard ADAM does not enjoy general convergence guarantees in the convex setting Reddi et al. (2018), although modifications such as AMSGRAD can address these issues. On the other hand, positive results have shown that under standard non-convex assumptions (smoothness, bounded gradients, and suitable stepsize decay), standard ADAM converges in expectation to a first-order stationary point with rate $\mathcal{O}(\log T/\sqrt{T})$ Chen et al. (2018); Zaheer et al. (2018). Our variant inherits these same guarantees: it converges under the same assumptions in the non-convex setting, and it inherits the same limitations in the convex setting. The analytical details are provided below.

**Notation and scaling.**  The notation in the main paper uses $\tilde{g}_t$ for the exponential moving average of the gradient, and includes certain correction factors. For the purpose of the convergence analysis, these factors can be absorbed into the learning rate schedule $\eta_t$, which does not affect the generality of the results. We also omit bias correction for clarity, since it only influences the constants in the first few iterations and does not affect asymptotic convergence guarantees (cf. Chen et al. (2018)). To simplify the presentation, we therefore write $m_t$ in place of $\tilde{g}_t$, assume that all multiplicative constants have been incorporated into $\eta_t$, and proceed with this cleaner notation in the proofs below.

We analyze the following update. For $t \geq 1$:

$$g_t = \nabla f(\theta_t), \tag{11}$$

$$m_t = \beta_1 m_{t-1} + (1 - \beta_1) g_t, \tag{12}$$

$$v_t = \beta_2 v_{t-1} + (1 - \beta_2) m_t^2, \tag{13}$$

$$\theta_{t+1} = \theta_t - \eta_t \frac{m_t}{\sqrt{v_t} + \epsilon}, \tag{14}$$

where squaring is elementwise and division is elementwise. We work under standard assumptions used in analyses of ADAM-type methods.

**Assumptions.**

(A1) (*Smoothness*) $f : \mathbb{R}^d \to \mathbb{R}$ is $L$-smooth:

$$f(y) \leq f(x) + \nabla f(x)^\top (y - x) + \tfrac{L}{2} \|y - x\|^2 \quad \forall x, y.$$

(A2) (*Bounded stochastic gradients*) $\mathbb{E}\|g_t\|^2 \leq G^2$ for all $t$.

(A3) (*Stepsizes / effective-steps summability*) Coordinate-wise,

$$\sum_{t=1}^{\infty} \frac{\eta_t}{\sqrt{v_{t,i}}} = \infty, \qquad \sum_{t=1}^{\infty} \frac{\eta_t^2}{\sqrt{v_{t,i}}} < \infty.$$

(A4) (*Numerical stability*) $\epsilon > 0$ is fixed, hence $\sqrt{v_{t,i}} + \epsilon \geq \epsilon$.

Assumption (A3) is the usual ADAM-type condition expressed directly in terms of the *effective* stepsizes (see, e.g., Chen et al. (2018); Zaheer et al. (2018)); it is satisfied by typical choices such as $\eta_t = \eta_0/\sqrt{t}$ under mild bounds on $v_t$.

### D.1  AUXILIARY BOUNDS

**Lemma D.1** (Bounded first moment). *Under (A2), for all $t$,*

$$\mathbb{E}\|m_t\|^2 \ \leq \ \frac{(1 - \beta_1)}{(1 + \beta_1)} \sum_{k=0}^{t-1} \beta_1^{2k} \, \mathbb{E}\|g_{t-k}\|^2 \ \leq \ \frac{1}{(1 - \beta_1)} \, G^2.$$

*In particular, each coordinate satisfies $\mathbb{E}[m_{t,i}^2] \leq G^2/(1 - \beta_1)$.*

*Proof.* Unroll $m_t = (1 - \beta_1) \sum_{k=0}^{t-1} \beta_1^k g_{t-k}$ and use Jensen/triangle and bounded second moment of $g_t$. The geometric-series bound yields the inequality.  $\square$

**Lemma D.2** (Bounds on the second-moment accumulator). *Define* $v_t = \beta_2 v_{t-1} + (1 - \beta_2)m_t^2$ *elementwise with any* $v_0 \geq 0$. *Then for each coordinate* $i$ *and all* $t$,

$$0 \leq \mathbb{E}[v_{t,i}] \leq \mathbb{E}[m_{t,i}^2] \leq \frac{G^2}{1 - \beta_1}.$$

*Consequently,* $\sqrt{v_{t,i}} \leq G/\sqrt{1 - \beta_1}$ *almost surely if* $g_t$ *is a.s. bounded, and in expectation otherwise. With (A4),* $\sqrt{v_{t,i}} + \epsilon$ *is bounded above and below by positive constants.*

*Proof.* Monotonicity of the EMA implies $v_{t,i} \leq \max_{s \leq t} m_{s,i}^2 \leq \sum_{s \leq t}(1 - \beta_2)\beta_2^{t-s} m_{s,i}^2$, and taking expectations with Lemma D.1 gives the stated bound. Nonnegativity is immediate from the recursion and $v_0 \geq 0$. $\qquad\square$

**Lemma D.3** (Effective steps remain controlled). *Under Lemma D.2 and (A4), there exist constants* $0 < c \leq C < \infty$ *such that for all coordinates* $i$,

$$c \leq \frac{1}{\sqrt{v_{t,i}} + \epsilon} \leq C, \qquad \forall t.$$

*Hence, if* $\{\eta_t\}$ *satisfies (A3), then the effective steps* $\eta_t/(\sqrt{v_{t,i}} + \epsilon)$ *satisfy the same summability properties.*

*Proof.* The bounds follow from the upper bound on $\sqrt{v_{t,i}}$ in Lemma D.2 and the lower bound $\epsilon$ in (A4). The summability statements are then immediate. $\qquad\square$

## D.2 DESCENT AND STATIONARITY

**Lemma D.4** (One-step descent). *Under (A1), for* $\Delta_t := \eta_t \frac{m_t}{\sqrt{v_t} + \epsilon}$,

$$f(\theta_{t+1}) \leq f(\theta_t) - \langle \nabla f(\theta_t), \Delta_t \rangle + \frac{L}{2}\|\Delta_t\|^2.$$

*Proof.* This is the standard smoothness (descent) inequality with $y = \theta_{t+1} = \theta_t - \Delta_t$. $\qquad\square$

**Theorem D.5** (Non-convex convergence to stationarity). *Assume (A1)–(A4). Then there is a constant* $C > 0$ *such that*

$$\min_{1 \leq t \leq T} \mathbb{E}\|\nabla f(\theta_t)\|^2 \leq C \frac{\log T}{\sqrt{T}}.$$

*Equivalently, our* ADAM *variant converges in expectation to a first-order stationary point with the same* $\tilde{\mathcal{O}}(1/\sqrt{T})$ *rate as standard* ADAM *(cf. Chen et al. (2018); Zaheer et al. (2018)).*

*Proof.* Following the template of ADAM-type analyses (e.g., Chen et al. (2018); Zaheer et al. (2018)), take expectations in Lemma D.4, sum over $t = 1, \ldots, T$, and use Cauchy–Schwarz on the correlation term:

$$\mathbb{E}\langle \nabla f(\theta_t), \Delta_t \rangle = \mathbb{E}\left[\sum_i \eta_t \frac{\nabla_i f(\theta_t) m_{t,i}}{\sqrt{v_{t,i}} + \epsilon}\right] \geq \kappa \eta_t \mathbb{E}\|\nabla f(\theta_t)\|^2 - \rho \eta_t \mathbb{E}\|m_t - \nabla f(\theta_t)\|^2,$$

for suitable constants $\kappa, \rho > 0$ depending on the bounds in Lemmas D.1–D.3. The variance term is handled as in Chen et al. (2018) by bounded gradients (A2) and EMA stability of $m_t$. The quadratic term $\frac{L}{2}\mathbb{E}\|\Delta_t\|^2$ is controlled by $\sum_t \eta_t^2/(\sqrt{v_{t,i}} + \epsilon)$, which is finite by Lemma D.3 and (A3). Telescoping the left-hand side yields

$$\sum_{t=1}^{T} \eta_t \mathbb{E}\|\nabla f(\theta_t)\|^2 \leq (f(\theta_1) - f^\star) + C_1 \sum_{t=1}^{T} \frac{\eta_t^2}{\sqrt{v_{t,i}} + \epsilon} + C_2 \sum_{t=1}^{T} \eta_t,$$

for constants $C_1, C_2$. With $\eta_t = \eta_0/\sqrt{t}$ and standard bounds on harmonic sums and $\sum_t \eta_t^2$, the claimed rate follows in the usual way (see, e.g., Chen et al. (2018); Zaheer et al. (2018) for the final aggregation). $\qquad\square$

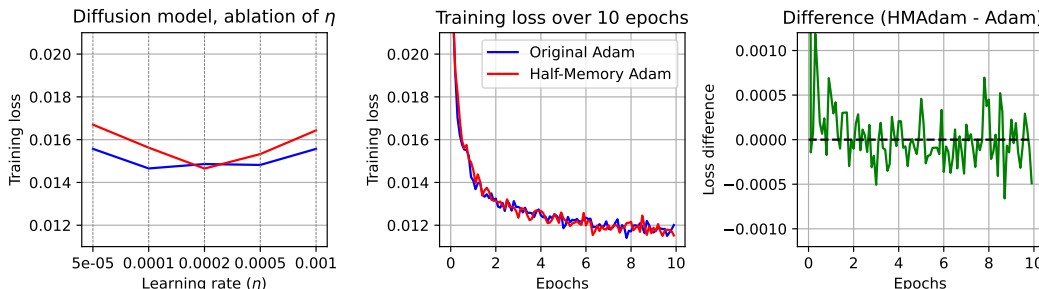

Figure 10: Comparison of ADAM and HALF-MEMORY ADAM on diffusion-based image generation with the CELEBA dataset. **Left:** Final training loss after 1 epoch (6,300 steps) across different learning rates, computed as the average over the last 1,000 steps. **Middle:** Training loss curves over 10 epochs using ($\eta = 0.0002$, $\beta_1 = 0.9$, $\beta_2 = 0.95$), showing highly similar trajectories. **Right:** Pointwise difference between the two optimizers, consistently below $10^{-3}$, indicating closely aligned performance throughout training.

# E    MORE EXPERIMENTS

## E.1    CELEBA IMAGE GENERATION

We further evaluate the two optimizers in the setting of generative modeling, where optimization stability is critical. Specifically, we consider diffusion-based image generation using the CelebA dataset (Liu et al., 2015), which contains 162,770 aligned celebrity face images. Diffusion models are an attractive benchmark because they require accurate tracking of gradient statistics across long training horizons, and are sensitive to optimizer stability.

For our experiments, we adopt the `minDiffusion` implementation (Lucidrains, 2021) with a lightweight UNet model containing approximately 450 K parameters, trained as a Denoising Diffusion Probabilistic Model (DDPM). Images are resized to $128 \times 128$, and training is performed using a standard mean-squared error (MSE) loss over 1,000 diffusion steps with a linear noise schedule. This setup is sufficient to highlight optimizer behavior without the computational burden of scaling to very large diffusion models.

We use a batch size of 32 and disable data augmentation. Both ADAMW and HMADAMW are configured with $\beta_1 = 0.9$ and $\beta_2 = 0.95$. We first conduct a one-epoch learning-rate sweep in the range $[5 \times 10^{-5}, 0.001]$, covering approximately 6,300 training steps. Based on this sweep, we select $\eta = 0.0002$ for further experiments and train both optimizers for 10 epochs under identical conditions.

Figure 10 presents the results. The left panel shows the learning-rate sweep, where both optimizers display nearly identical sensitivity curves. The middle panel shows training loss trajectories during the 10-epoch runs, which overlap almost perfectly. The right panel plots the pointwise difference between losses, which remains consistently below $10^{-3}$ throughout training. These results indicate that HMADAMW preserves the optimization dynamics of ADAMW even in a sensitive generative modeling task, while requiring half the optimizer state memory.

These experiments are not intended to compete with state-of-the-art diffusion training, but to confirm that HMADAMW matches the behavior of ADAMW on a generative vision task under controlled conditions.

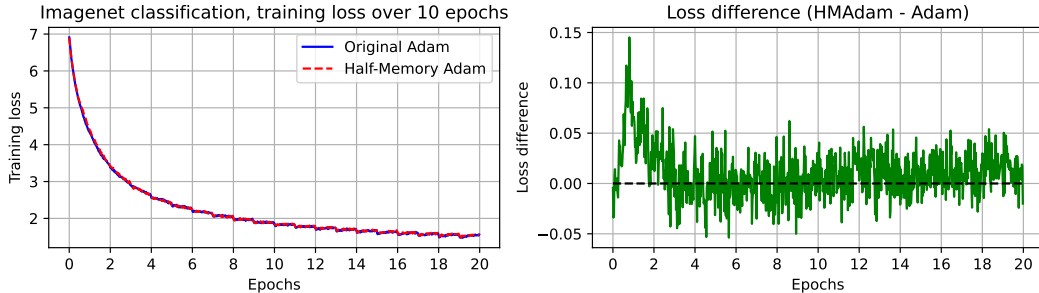

Figure 11: Comparison of ADAM and HALF-MEMORY ADAM on ImageNet classification using an EFFICIENTNET-B7 model. The model is trained with fixed hyperparameters: learning rate $\eta = 0.0002$, $\beta_1 = 0.9$, and $\beta_2 = 0.99$. **Left:** Training loss over 20 epochs. The two optimizers exhibit nearly identical learning trajectories. **Right:** Pointwise difference between the training losses of the two optimizers, which remains mostly below 0.1 throughout training.

### E.2 IMAGENET CLASSIFICATION

To evaluate our optimizer on a high-capacity convolutional model, we test EfficientNet-B7 (Tan & Le, 2019), which has approximately 66 M parameters. Although EfficientNet is typically trained with RMSProp, here we deliberately use ADAMW for both baselines to ensure comparability across tasks and optimizers. This isolates the effect of optimizer state size rather than confounding optimizer choice.

We use fixed hyperparameters ($\eta = 0.0002$, $\beta_1 = 0.9$, $\beta_2 = 0.99$, weight decay 0.01), train for 20 epochs on ImageNet (ILSVRC 2012) (Deng et al., 2009) with a batch size of 32, and disable data augmentation. Results are shown in Figure E.2: both optimizers exhibit nearly identical training trajectories, with loss differences typically below 0.1. This indicates that given the same starting point and hyperparameters, their behavior is statistically indistinguishable.

In terms of memory usage, the difference between the two optimizers matches theoretical predictions. For EfficientNet-B7, the baseline ADAMW required approximately 532 MB of optimizer state, while HMADAMW used only 266 MB. This confirms that one of the two auxiliary states can be eliminated without altering convergence behavior.

We also measured runtime efficiency. Despite halving optimizer memory, HMADAMW incurred no computational overhead. Both optimizers averaged 26 ms per optimization step, with no measurable difference in throughput. This demonstrates that the modification preserves not only the convergence properties of ADAMW but also its runtime characteristics.

These ImageNet experiments are supplementary and not intended to reach state-of-the-art accuracy. Their purpose is to demonstrate that HMADAMW reliably tracks the training dynamics of ADAMW in a convolutional setting while reducing optimizer state memory by half. While we report 20-epoch results here for efficiency, we plan to extend them to the standard 90 epochs by the camera-ready version. Importantly, our conclusions do not depend on absolute accuracy: the key result is that the two optimizers remain aligned and statistically indistinguishable under matched hyperparameters.

