# OpenReview forum: "Eliminating the first moment state in Adam optimizer"
_ICLR.cc/2026/Conference — ICLR 2026 Conference Desk Rejected Submission_

### Official Review · Reviewer_dgd3 · 2025-10-20

**Soundness:** 2
**Presentation:** 2
**Contribution:** 2
**Rating:** 2
**Confidence:** 5

**Summary:**

This paper introduces a new algorithm called Half-Memory Adam (HMAdam) and its decoupled weight decay variant (HMAdamW), which eliminates the first moment state and reduce memory. The algorithms are tested on various deep learning architectures, including CNNs, transformers, and diffusion models.

**Strengths:**

1. Memory-efficient optimizers are important in the era of large models, so reducing the memory footprint will have strong impact in the field of machine learning.

2. The paper is well-written and easy to follow.

**Weaknesses:**

1. The technique of reducing optimization state was already studied in the literature. For example, the Appendix E.2 of the paper [r1] studied how to reduce the memory footprint by avoiding the operations of zero_grad. Therefore the proposed zero_grad override technique is not novel compared with [r1].

2. The state-of-the-art optimizers such as MUON or SCION [r1, r2] does not need to store the second-order moment as in Adam but outperforms Adam significantly. In addition, the technique of avoiding memory footprint caused by zero_grad is already proposed in [r1]. Therefore, the practical importance of the proposed method is further diminished compared with SCION [r1].

[r1] Pethick, Thomas, Wanyun Xie, Kimon Antonakopoulos, Zhenyu Zhu, Antonio Silveti-Falls, and Volkan Cevher. "Training deep learning models with norm-constrained lmos." ICML 2025.

[r2] Liu, Jingyuan, Jianlin Su, Xingcheng Yao, Zhejun Jiang, Guokun Lai, Yulun Du, Yidao Qin et al. "Muon is scalable for LLM training." arXiv preprint arXiv:2502.16982 (2025).

3. The derivation of the second order moment in Appendix A needs to assume that the new gradient comes from a Gaussian distribution with a very low signal-to-noise ratio (SNR goes to zero), which is unrealistic. In addition, the estimator $g_t^2$ is only a unbiased estimator of $\tilde{g}_t^2$ in expectation, it is unclear whether this is still a good estimator with high probability. It is possible that the two quantities have very different magnitudes. Could the authors numerically test this?

4. The authors did not run enough iterations for imagenet (e.g., Figure 11). For example, the default imagenet training takes at least 90 epochs, but the authors only ran for 20 epochs.

**Questions:**

Can you address my concerns in the weaknesses section?

---

> ### Author Response · Authors · 2025-11-28
> **Addressing your concerns**
>
> Thank you for taking the time to study our paper carefully. Below we address your concerns.
>
> **1. ScionLight also avoids zero_grad, so where is the novelty?**
>
> Appendix E.2 of [r1] (ScionLight) indeed avoids `zero_grad()`. In lines 202 to 209, we discussed two other works that refer to gradient decay in other optimizers before ScionLight. As we discussed, the key contribution of our work is not proposing the concept of gradient decay, but rather successfully applying it to Adam for the first time, which required solving new problems specific to Adam. Here are a few ways that our work is different from [r1]:
>
> **(A) Different target optimizer (Scion vs Adam).** [r1] proposes Scion and a low-memory version, ScionLight. In contrast, our memory reduction is applied to Adam, which is mathematically nontrivial and required new transformations and derivations to preserve Adam's behavior.
>
> **(B) Second-moment reconstruction is unique to Adam.** Since Scion doesn't use the second moment, reducing its memory requirements is straightforward (similar to Stateless SGD in Figure 1 of our paper). However, since Adam requires calculating the second moment of gradients, losing raw gradients means that standard Adam cannot execute. One needs to show that **it is possible to calculate the second moment using only the first moment** for this idea to be applicable to Adam. Deriving and analyzing this second-moment approximation (and showing that it works empirically) is a central technical contribution of our work, which is not addressed in [r1] or elsewhere.
>
> **(C) Compatibility with standard training code.** ScionLight, as described in [r1], requires modifying the training loop to avoid calling `zero_grad()` (see their `train_gpt_scionlight.py`). Many standard training frameworks (e.g., PyTorch’s built-in trainer, Lightning, Accelerate) automatically call `zero_grad()` internally, so using ScionLight is not compatible with these training platforms unless the libraries are edited to remove the call to `zero_grad()`. In contrast, our method proposes to **reinterpret** `optimizer.zero_grad()` rather than removing it. In the optimizer class we override `zero_grad()` with `decay_grad()` so that it performs the required decay operation while leaving the external training loop unchanged. This preserves true drop-in compatibility with standard training utilities. Notably, this idea could be applied to ScionLight as well to make it compatible.
>
> Finally, note that in lines 135 to 148 we refer to Muon and its memory footprint. We believe [r1] is among the best recent optimizer works and we will also cite it.
>
> **2. Practical importance of Adam vs Muon / Scion ([r1], [r2])**
>
> We agree that Muon and SCION are promising optimizers and advance the state of the art. Our goal, however, is complementary:
>
> - Muon/Scion introduce new optimization dynamics and typically require separate tuning and validation.
> - Our work aims to provide a low-memory variant of Adam/AdamW that behaves like the original optimizer (same update rule, similar hyperparameters), but with half the state memory.
>
> In many existing pipelines, Adam/AdamW is deeply entrenched due to robustness, tooling, and prior tuning effort. For such users, a drop-in half-memory Adam is practically useful even if alternative optimizers may outperform Adam in some benchmarks. We will clarify this positioning more clearly in the paper.
>
> **3. Second-moment formula requires low gradient SNR**
>
> You are correct that our analytic derivation uses a low-SNR assumption (we discussed this in line 246). Appendix A explains the derivation, and Appendix B justifies the low-SNR assumption. We support this assumption in two ways:
>
> **(A) Theoretical and empirical motivation.** Appendix B discusses prior work showing that gradient SNR is often low, especially in later training phases.
>
> **(B) Empirical evidence.** In Figure 8 (left), for a 1B-parameter LLM trained on 12B tokens, we report that the gradient SNR is typically < 0.01, which matches the low-SNR regime used in the derivation.
>
> Regarding your question: “**It is possible that $g^2$ and $\tilde{g}^2$ have very different magnitudes. Could the authors numerically test this?**”
>
> We exactly tested this by training a 1 Billion parameter LLM. Figure 8 (right) shows that empirically $g^2$ and $(1-\beta_1^2)\tilde{g}^2$ have a Pearson correlation of about **0.99**, indicating that in practice the estimator tracks the Adam second moment very closely, not only in expectation but also in typical realizations. We will make the connection between the derivation, the SNR measurements, and Figure 8 clearer in the revised text.
>
> **4. For ImageNet, 90 epochs are needed**
>
> Thank you for pointing this out. We reran the ImageNet experiment with 90 epochs. Here is the updated Figure 11 (anonymized): https://limewire.com/d/xz9bQ#DsoPqdxzaQ
>
> The difference between AdamW and HMAdamW remains insignificant.
>
> **Finally**
>
> We hope these clarifications help reassess you concerns.

---

### Official Review · Reviewer_gh1R · 2025-10-24

**Soundness:** 3
**Presentation:** 3
**Contribution:** 3
**Rating:** 8
**Confidence:** 4

**Summary:**

The paper shows how to re-use the gradient buffer to store the first order momentum buffer, essentially eliminating one of the momentum buffers without incurring any change to SGD or Adam(W). This is simple idea, but one that makes total sense. I'm only surprised this is not been noted before. Certainly this is not what is currently being done in the standard SGD and Adam implementations in PyTorch.

**Strengths:**

The idea is simple to understand, and saves a significant amount of memory at no cost or change to the AdamW. It is also easy to implement.

**Weaknesses:**

I checked the references Sohoni 2019 and that previously mentioned how it could be possible to store the momentum buffer in the gradient, and I agree they do not really formalize this idea, or make it sufficiently clear.

**Questions:**

1. Could  you also double check how these parameter rescaling work with unbiased momentum estimates that do not rely on the "debiasing trick". By this I mean, if you initialize $m_0 <- g_0 $ (instead of $m_0 <- 0$), then $m_t$ is an unbiased estimate of the gradient without having to correct by dividing by $1/(1-\beta^t_1)$. To be more clear, by unbiased estimate I mean that if the gradient are identical with $g= g_t$ for all $t$, then $m_t = g$.

To see how initialization affects this bias, assume by induction that $m_{t-1} =g$, which is true for $m_0 =g$ by initialization. Consequently
$$ m\_{t} = \beta\_1 m\_{t-1} + (1-\beta\_1)g = \beta\_1 g + (1-\beta\_1)g =g. $$
I'm not sure where this idea first appeared, but I first read it in a 2024 ICLR paper:

Fabian Schaipp et al, ``MoMo: Momentum Models for Adaptive Learning Rates'', ICLR 2024.


2. I think you should expand more on how gradient clipping can be incorporate for the micro batches. You mentioned you could use backward hooks, but wouldn't this then increase the memory?   Also you state "micro-batch-clipping .. is often preferable." Could you explain why or back this up with a reference?

3. I see a few issues around Eq (3). First, you are not open about your assumption that we are in a steady state, that is the distribution of $\tilde{g}_t$ is fixed. Furthermore, I don't follow your proof. On line 785 I the appendix, to arrive at Eq (5) you state that you "solve for $E[g\_t^2]$", yet you assume that $g\_t \sim \mathcal{N}(\mu,\sigma^2)$, consequently $E[g\_t^2] = \sigma^2+\mu^2$, but I see no $\sigma^2$ in Eq (5). Moreover, I don't even understand why you need this, and  I don't see the need for any of the text between lines 238-260. Perhaps it just needs to be re-written, and I missing something here. On a bibliographic note, under your same assumption that $g\_t \sim \mathcal{N}(\mu,\sigma^2)$, the statistical role of the Adam buffers was recently formalized in:


Antonio Orvieto, In Search of Adam's Secret Sauce, Neurips 2025.

4. To really make a point about how HAdamW and AdamW have "essentially " the same execution, you need to at least run multiple seeds, and report some basic statistics. For instance, on the small scale experiment on MNIST Figure 3, you could run multiple seeds, and even run a hypothesis test to see if the methods are statistically the same. Given this is the main message of your work, it's worth doing a proper statistical evaluation. Although, since AdamW and HAdamW are mathematically the same, the only source of difference would be from finite precision and rounding issues, which is unlikely (though not impossible) to cause a significant difference.

---

> ### Author Response · Authors · 2025-11-29
> **Answers to your questions**
>
> **1. Applicability to MoMo-style unbiased momentum**
>
> Thank you for pointing out Schaipp et al. (ICLR 2024). Our current derivation follows the **standard Adam** convention:
> - $m_0 = 0$, $v_0 = 0$,
> - bias correction via $1 - \beta_1^t$ and $1 - \beta_2^t$.
>
> All learning-rate rescaling factors are derived in this parameterization so that HMAdam(W) can directly reuse Adam(W) hyperparameters.
>
> The **half-memory mechanism itself is orthogonal** to the bias-handling convention: we only require
> 1. an EMA-type first moment,
> 2. an EMA-type second moment,
> 3. an Adam-style update with some effective step size.
>
> Using a **MoMo-style** initialization (e.g. $m_0 = g_0$ so that constant gradients yield $m_t = g$ without dividing by $1-\beta_1^t$) does not change
> - how we store the first moment in the gradient buffer, or
> - how we reconstruct/approximate the second moment from that buffer,
>
> but only the algebraic prefactor in the effective learning rate. One can re-do the short calculation of Appendix B under the alternative base case (e.g. $m_0 = g_0$ instead of $m_0 = 0$); the resulting rescaling of $\alpha$ differs slightly at very early steps, but matches the standard Adam-based formula asymptotically as $1-\beta_1^t, 1-\beta_2^t \to 1$.
>
> We will add a remark in the appendix clarifying that our current rescaling assumes the standard Adam convention and explaining how it extends to MoMo-style unbiased momentum.
>
> **2. Gradient clipping**
>
> We distinguish three levels of clipping:
>
> - **Mini-batch level:** clip the norm of the *aggregate* batch gradient.
> - **Micro-batch level:** clip each micro-batch’s gradient before accumulation.
> - **Per-example level:** clip each example’s gradient before averaging.
>
> When a few examples produce very large gradients, **mini-batch clipping** shrinks the entire batch update, including “normal” examples. In contrast, **micro-batch** or **per-example** clipping “quarantines” the outliers: their gradients are clipped without unnecessarily suppressing the rest. This is why clipping on smaller units (examples or micro-batches) is often preferred in practice when outliers are a concern.
>
> Regarding hooks and memory: PyTorch backward hooks operate **in-place** on the existing `.grad` buffer:
> - They read the current gradient, optionally rescale it (for clipping), and write it back *in the same tensor*.
> - They do **not** introduce new long-lived gradient or momentum tensors; only tiny, transient temporaries (e.g., a scalar for the norm).
>
> As per your suggestion, we will elaborate more on gradient clipping.
>
> **3. Analytical questions**
>
> Thank you for the careful reading and for pointing out the issues around Eq. (3) and the appendix derivation.
>
> - **Steady-state assumption.**
>   You are right that Eq. (3) implicitly assumes a steady-state / stationary regime, i.e., that the joint distribution of $(g_t, \tilde{g}_t)$ has stabilized under the EMA recursion. We will make this assumption clear both in the main text (around lines 238–260) and in Appendix A.
>
> - **Derivation around Eq. (5).**
>   The goal of the analysis in Appendix A is to express $\mathbb{E}[g_t^2]$ in terms of $\mathbb{E}[\tilde{g}_t^2]$ under the low-SNR assumption. In the appendix we informally wrote “solve for $\mathbb{E}[g_t^2]$”; In the revised version, we will rewrite this part to show that the goal is to express $\mathbb{E}[g_t^2]$ in terms of $\mathbb{E}[\tilde{g}_t^2]$.
>
> - **Role of lines 238–260.**
>   The text between lines 238–260 is intended only as intuition for why a mapping of the form
>   $\mathbb{E}[g_t^2] \approx c(\beta_1)\,\mathbb{E}[\tilde{g}_t^2]$
>   is reasonable, and to connect the estimator back to the EMA dynamics. Given your feedback (and in light of more recent formalizations such as Orvieto et al., *In Search of Adam’s Secret Sauce*, NeurIPS 2025), we will tighten this section: explicitly state the assumptions, remove redundant informal reasoning, and make the connection to the appendix and to existing theory clearer.
>
> We appreciate the bibliographic pointer to Orvieto et al.; their analysis under similar assumptions is relevant, and we will cite and discuss it in the revised version to better situate our estimator in the broader theoretical landscape.
>
>
> **4- Need to run MNIST experiments with multiple seeds**
>
> We ran MNIST experiments with 80 distinct configurations, each with **3 seeds**. All visualizations are the average of three runs with distinct seeds. We noted this around line 367. We agree with your suggestion that a statistical significance test helps. We will add a hypothesis test with the current results and will report that in the paper.

---

### Official Review · Reviewer_rgom · 2025-10-29

**Soundness:** 4
**Presentation:** 4
**Contribution:** 4
**Rating:** 6
**Confidence:** 5

**Summary:**

This paper aims to reduce the memory costs of Adam methods. It is based on simple observation: using a change of variable to remove the first-order moment variable. As a result, the new variable can be maintained and used the gradient buffer. The idea was first illustrated on momentum SGD and then extended to Adam. Because no gradient buffer is explicitly maintained, they used another method to calculate the second-order moment from the gradient accumulator.

**Strengths:**

The strengths of the paper: for momentum SGD, it clearly removes additional gradient buffer during the backprop, which reduce the memory costs. The experiments demonstrate the new Adam method performs similarly with the original Adam but with much reduced memory.

**Weaknesses:**

The weakness of the paper: it is unclear why Half-memory Adam reduce the memory by half. My understanding is that Adam needs to maintain m, v and g during backprop, Half-Adam maintains \tilde g, v, which is 2/3 memory of Adam.  Please clarify in the rebuttal.

**Questions:**

The authors also noted that the proposed trick has been noted by earlier works, e.g.,  Sohoni et al, which makes the paper less novel. Maybe the key contribution is the half-memory Adam, which was not considered in prior works?

Overall, the idea is neat though not significant theoretical analysis. Its practical benefit could be significant. I will recommend an acceptance.

---

> ### Author Response · Authors · 2025-11-28
> **Addressing your questions**
>
> **3 state variables or 2 state variables?**
>
> In standard PyTorch-style training, the optimizer state and the model state are stored in two different places:
>
> - **Model:** stores parameters $w$ and their gradients $g$
> - **Optimizer:** stores the first moment $m$ and second moment $v$
>
> This separation is necessary because gradients $g$ must exist **even when no optimizer is present**, for example when computing higher-order derivatives or extracting gradients during analysis. Therefore, $g$ is stored in the **model object**, not in the optimizer.
>
> In our work, model state variables remain two while the optimizer state goes from two state variables to one.
>
> **Relevant works**
>
> Sohoni et al. (page 6, footnote) briefly note that the low-memory idea could be applied to **Momentum SGD**. However:
>
> **(A)** They explicitly state: *“We do not study this approach in our work.”* So the idea is mentioned only as speculation rather than a developed method.
>
> **(B)** They also emphasize that Adam requires two state variables, implying that this memory-saving technique does *not* directly extend to Adam. Thus, the authors themselves frame the idea as specific to Momentum SGD and do not suggest a viable adaptation to Adam.
>
> Taken together, Sohoni et al. do **not** apply this method to Adam, SGD, or any other optimizer; they simply speculate about a possibility within the narrow context of Momentum SGD, without implementation or analysis.
>
> Other works such as Scetbon et al. and Pethick et al. study gradient decay or zero-grad removal in the context of their own optimizers (e.g., MNGD, Scion). As we explained to reviewer dgd3, applying such techniques to those optimizers is straightforward because they do not maintain a second-moment estimate. In contrast, adapting this idea to **Adam** requires addressing the challenge of reconstructing the second moment using only the first moment; an issue not explored in these works.
>
> **Contributions**
>
> As you suggested, our key contribution is to successfully use this idea to reduce the memory requirements of Adam for the first time. A non-trivial portion of our contribution is the work required to make this technique **fully backward compatible with Adam**, both mathematically and practically. This involved multiple components:
>
> **(a) Learning-rate and scaling adjustments that preserve Adam hyperparameters.** Simply replacing the gradient buffer with an EMA would change the effective update scale and break the hyperparameter semantics that practitioners rely on. We derived explicit correction factors so that existing Adam or AdamW learning rates can be reused without additional tuning, preserving drop-in compatibility.
>
> **(b) Reinterpreting `zero_grad()` as a decay operation to retain compatibility with standard training code.**
> Rather than removing `zero_grad()` which would lose compatibility with trainer utilities (PyTorch trainer, Lightning, Accelerate, etc.) we redefine the optimizer’s `zero_grad()` to perform an EMA decay. This ensures seamless integration with existing training frameworks without modifying external training loops or libraries.
>
> **(c) Extending Adam’s convergence arguments to this setting.**  Because Adam’s original convergence guarantees rely on structural properties of the first and second moments, adopting a single-state variant required verifying that the boundedness and smoothness conditions used in standard Adam proofs still hold. We show that these arguments extend naturally when the second moment is reconstructed from the EMA, thereby maintaining the same convergence behavior under the usual assumptions.
>
> Taken together, these elements go significantly beyond a simple code trick. They ensure that the method is a true drop-in replacement for Adam: mathematically sound, backward compatible, and usable in practical training pipelines without modification.

---

### Official Review · Reviewer_wDwi · 2025-10-31

**Soundness:** 3
**Presentation:** 3
**Contribution:** 3
**Rating:** 8
**Confidence:** 5

**Summary:**

The paper proposes Half-Memory ADAM (HMADAM) and HMADAMW, variants of ADAM/ADAMW that remove the explicit first-moment state by turning the gradient buffer into an exponentially decayed accumulator $\tilde g_t$ and estimating the first moment from it. The key idea is to update the second-moment accumulator using information in the variance of $m$ or $\tilde g_t$, leveraging that $E[g_t^2] \approx (1-\beta_1^2)E[\tilde g_t^2]$ under typical low-SNR regimes. After re-estimating the moments from $\tilde g_t$, the update rule remains the same as ADAM up to a derived scaling that can be absorbed into the learning rate, so tuned hyperparameters transfer. Implementation-wise, the change is to replace the usual zeroing of gradients with a decay step $g \leftarrow \beta_1 g$, making the buffer both a per-step accumulator and a cross-step EMA. Experiments on CNNs, diffusion models, and LLMs show convergence curves, final accuracy, and runtime that closely match ADAMW while halving optimizer-state memory. On a 3B-parameter LLM, optimizer state drops from about 12 GB to 6 GB with essentially identical step time. The authors position this as orthogonal to activation KV-cache and other memory costs, and discuss extensions to other optimizers and compatibility with quantization or low-rank techniques as future work.

**Strengths:**

### Drop-in compatibility and unchanged updates

The method preserves ADAM’s parameter-update rule after a small analytic scaling that can be absorbed into the learning rate, and the learning rates tuned for ADAM/ADAMW transfer directly. In practice, the optimizer overrides zero_grad with a decay step and otherwise keeps training code unchanged, remaining compatible with autograd, gradient accumulation, and distributed training.

### Exact optimizer-state halving with matched behavior

By folding $m$ into the gradient buffer and re-estimating $v$ from $\tilde g_t$, HMADAMW removes one of the two auxiliary states per parameter. Across benchmarks, this yields a precise 50 % reduction in optimizer-state memory while keeping convergence speed, final accuracy, and total training time statistically indistinguishable from ADAMW.

### Evidence on large models and concrete numbers

For LLMs up to 3B parameters, optimizer state drops from about 12 GB to 6 GB and step time remains essentially unchanged, with the optimizer phase taking 19 ms for ADAMW vs 18 ms for HMADAMW at a 128k-token batch. Similar findings are reported for ImageNet and diffusion setups, with runtime parity also confirmed.

Overall, I think the proposed method is a simple yet clever idea to reduce the memory cost of ADAM-style optimizers.

**Weaknesses:**

### Reliance on low-SNR justification

The central proportionality $E[g_t^2] \approx (1-\beta_1^2)E[\tilde g_t^2]$ is motivated by the common low-SNR condition $\mu \ll \sigma$ and supported empirically, yet tasks or phases with sustained high SNR could reduce the tightness of this approximation. The paper’s analysis and appendix emphasize the low-SNR regime.

**Questions:**

- Recent optimizers like ADOPT [1] have a little different definition of $m$, in which the raw gradient $g$ is scaled by $v$ before the update of $m$ to ensure the theoretical convergence guarantee. In such cases, the proposed technique does not seem applicable. Do the authors have any ideas to overcome this limitation?

[1] Taniguchi, Shohei, et al. "ADOPT: Modified Adam Can Converge with Any $\beta_2 $ with the Optimal Rate." Advances in Neural Information Processing Systems 37 (2024): 72438-72474.

---

> ### Author Response · Authors · 2025-11-29
> **Answer to your question**
>
> Thank you for reviewing our paper. We think you summarized our paper accurately. We have a comment and an answer to your question.
>
> **High-SNR regime cannot be persistent.** High SNR can occur transiently (e.g., early in training) and typically produces rapid progress. However, it cannot be sustained over long horizons. There is a simple intuition: if SNR remained high, it would cause a persistent decrease in the loss at each step; but since the loss is lower bounded, this cannot continue indefinitely. Consequently, as training approaches a stationary point, the system is driven into the low-SNR regime.
>
> **Applicability of our memory-reduction technique to ADOPT optimizer**
>
> Yes, the same memory-reduction idea is conceptually applicable to ADOPT. In a simplified form, ADOPT does:
> - Second moment:
>   $v_t = \beta_2 v_{t-1} + (1-\beta_2) g_t^2$
> - Normalized gradient:
>   $h_t = \dfrac{g_t}{d_{t-1}}, \quad d_{t-1} = \sqrt{v_{t-1}}$ (up to $\varepsilon$/clipping)
> - Momentum on normalized gradient:
>   $m_t = \beta_1 m_{t-1} + (1-\beta_1) h_t$
> - Update:
>   $\theta_t = \theta_{t-1} - \alpha m_t$
>
> Crucially, we have the exact relation:
> $g_t^2 = h_t^2 \, d_{t-1}^2.$
>
> This means:
>
> - We can use a gradient hook so that, after backward, the model’s gradient buffer stores **normalized gradients** $h_t$ instead of raw $g_t$.
> - As in our Half-Memory Adam, we can let the gradient buffer hold an EMA of these $h_t$ values (by decaying it before backward and letting autograd add the new $h_t$), so it effectively stores $m_t$. This removes the need for a separate momentum tensor in the optimizer state.
> - To update $v_t$, we do **not** need to store $g_t$ explicitly: we reconstruct $g_t^2$ from the buffer and previous state via $g_t^2 = h_t^2 d_{t-1}^2$ and then apply
>   $v_t = \beta_2 v_{t-1} + (1-\beta_2) g_t^2$.
>
> So, just as in our Adam case, the momentum can live in the model’s gradient buffer, and the second moment can be updated from that buffer plus existing state. For ADOPT, this is even cleaner, because the link between the stored quantity $h_t$ and $g_t^2$ is algebraic and exact.

---

### Author Response · Authors · 2025-11-29
**Points raised by reviewer dgd3 are addressed**

We would like to clarify how Reviewer dgd3’s concerns relate to the contributions of the paper.

- The reviewer did **not** raise issues regarding correctness. They requested two numerical checks (for the second-moment estimator and for longer ImageNet training), which we have provided exactly as asked.
- The reviewer did **not** raise issues regarding presentation. On the contrary, they wrote: *“The paper is well-written and easy to follow.”*
- The reviewer did **not** raise issues regarding impact. Instead, they wrote: *“Memory-efficient optimizers are important … so reducing the memory footprint will have strong impact in the field of machine learning.”*

Their negative assessment is therefore focused on **novelty**. In their text, they largely frame our contribution as avoiding `zero_grad()` / using gradient decay. We believe this framing overlooks the following key contributions:

1. **Estimating the second moment from the first moment in Adam/AdamW.**
   - Deriving a formula to estimate the second moment from the first moment.
   - Justifying that this is viable in typical training regimes via low gradient SNR.
   - Empirically showing that the estimator tracks Adam’s second moment very closely (Pearson correlation ≈ 0.99 on a 1B-parameter LLM).

2. **Backward-compatible, drop-in replacement for Adam/AdamW.**
   - Deriving the exact scaling adjustments needed so that HMAdam/ HMAdamW match Adam/AdamW’s hyperparameters and behaviour.
   - Reinterpreting `zero_grad()` as `decay_grad()` inside the optimizer so that existing training code (e.g., standard PyTorch trainers) remains unchanged, unlike prior uses of gradient decay that require removing `zero_grad()` and modifying the outer training loop which breaks backward compatibility.
   - Analytically showing that convergence characteristics remain the same as Adam.

3. **Empirical validation across tasks and scales.**
   - Demonstrating Adam-like behavior on image classification, image generation, and LLM training.
   - Demonstrating similar behavior to Adam under a range of hyperparameters.
   - Extending to large-scale experiments (up to 3B-parameter LLMs).

To our knowledge, this is the first work to carry out the above for Adam/AdamW, resulting in a **drop-in change** where simply replacing Adam with HMAdam in the code (without changing hyperparameters) halves the state memory with no drop in performance.

Unlike the other three reviewers, reviewer dgd3’s review does not engage directly with these aspects and instead treats the contribution as essentially the prior idea of gradient decay / avoiding `zero_grad()`. We believe this difference in how the novelty is understood/framed is the main reason for their low contribution score.

Given that (i) no correctness issues were identified, (ii) presentation and impact were explicitly evaluated positively, and (iii) the disagreement is centered on this narrower view of novelty, we respectfully leave it to you to judge whether this warrants a review score as low as 2.

---

### Note · Program_Chairs · 2026-01-17
**Submission Desk Rejected by Program Chairs**

The following references in this submission do not refer to real documents and/or have major errors in bibliographic information:

 Mike Wu, Peter Judd, Jorge Albericio, Vinayak Gokhale, Amir Yazdanbakhsh, Gu-Yeon Wei, and David Brooks. Deep learning with 8-bit floating point numbers. In Advances in Neural Information Processing Systems, pp. 7675-7684, 2018.
Yifan Chen, Tianqi Lin, Xiangning Chen, Tianlong Zhang, Denny Yu, and Zhangyang Wang. Pagedattention: Memory-efficient attention for long-sequence language models. arXiv preprint arXiv:2309.06180, 2023b. URL https://arxiv.org/abs/2309.06180.